# Knockout of the longevity gene Klotho perturbs aging and Alzheimer's disease-linked brain microRNAs and tRNA fragments

Serafima Dubnov [1], Estelle R. Bennett[2], Nadav Yayon [1,10,11], Or Yakov[2], David A. Bennett[3], Sudha Seshadri [4], Elliott Mufson[5], Yonat Tzur[2], David Greenberg[2], Makoto Kuro-o [6], Iddo Paldor[2,7], Carmela R. Abraham [8,9] & Hermona Soreq [1,2] ✉

Overexpression of the longevity gene Klotho prolongs lifespan, while its knockout shortens lifespan and impairs cognition via perturbation of myelination and synapse formation. However, comprehensive analysis of Klotho knockout effects on mammalian brain transcriptomics is lacking. Here, we report that Klotho knockout alters the levels of aging- and cognition related mRNAs, long non-coding RNAs, microRNAs and tRNA fragments. These include altered neuronal and glial regulators in murine models of aging and Alzheimer's disease and in human Alzheimer's disease post-mortem brains. We further demonstrate interaction of the knockout-elevated tRNA fragments with the spliceosome, possibly affecting RNA processing. Last, we present cell type-specific short RNA-seq datasets from FACS-sorted neurons and microglia of live human brain tissue demonstrating in-depth cell-type association of Klotho knockout-perturbed microRNAs. Together, our findings reveal multiple RNA transcripts in both neurons and glia from murine and human brain that are perturbed in Klotho deficiency and are aging- and neurodegeneration-related.

Age is a key risk factor for many neurodegenerative diseases[1], and identifying transcripts distinguishing between healthy aging and age-associated cognitive decline remains a major research goal. The Klotho protein exhibits remarkable properties of lifespan regulation[2], which has attracted significant interest in aging research for over two decades. Klotho-deficient mice show premature aging symptoms, including cognitive decline[3], whereas Klotho overexpression improves amyloid-β clearance and cognition in one Alzheimer's disease (AD) mouse model[4], and long-term potentiation (LTP) and behavior in another[5]. Klotho excess also improved remyelination, decreased motor neuron death and lowered neuroinflammation in a mouse model of multiple sclerosis[6,7]. Furthermore, injected Klotho enhances cognition in aged rhesus macaques[8], while elevated levels of functional Klotho

protein in humans is associated with both longevity[9] and lower AD risk[10]. On the contrary, human AD patients show lower Klotho levels in the cerebrospinal fluid (CSF) than age-matched controls[11]. Thus, the neuroprotective effects of Klotho are evolutionary conserved and functional in the mammalian brain.

While Klotho's involvement is well recognized in multiple brain processes, including myelination and synaptic plasticity[5,12], the impact of Klotho deficiency on both coding and non-coding RNA transcriptomes is incompletely understood. To address this issue, we compared short and long RNA-sequencing profiles of brain tissues from a Klotho knockout (KO) murine model to those of wild-type (WT) mice (n = 5 in each group), and identified changes in numerous RNA fragments. We sought correlations of

[1]The Edmond & Lily Safra Center for Brain Sciences, The Hebrew University of Jerusalem, 9190401 Jerusalem, Israel. [2]The Alexander Silberman Institute of Life Sciences, The Hebrew University of Jerusalem, 9190401 Jerusalem, Israel. [3]Rush Alzheimer's Disease Center, Rush University Medical Center, Chicago, IL, USA. [4]UT Health Medical Arts & Research Center, San Antonio, TX, USA. [5]Dept. Translational Neuroscience, Barrow Neurological Institute, St. Joseph's Medical Center, Phoenix, AZ, USA. [6]Division of Anti-aging Medicine, Center for Molecular Medicine, Jichi Medical University, Shimotsuke, Tochigi, Japan. [7]Dept of Neurosurgery, the Shaare Zedek Medical Center, Jerusalem, Israel. [8]Departments of Biochemistry and Pharmacology & Experimental Therapeutics, Boston University School of Medicine, Boston, MA, USA. [9]Klogenix LLC., Boston, MA, USA. [10]Present address: Wellcome Sanger Institute, Wellcome Genome Campus, Hinxton, Cambridge, UK. [11]Present address: European Molecular Biology Laboratory European Bioinformatics Institute, Hinxton, Cambridge, UK. ✉e-mail: hermona.soreq@mail.huji.ac.il

the observed transcriptomic changes with reported aging and AD alterations, and pursued the cell type-specificity of the observed differences. Among short RNAs, we focused on microRNAs (miRs) and transfer RNA fragments (tRFs), due to their suggested involvement in brain aging and neurodegeneration[13–15]. We further established that the tRF upregulated by Klotho knockout interacts with proteins involved in major RNA processing pathways, such as splicing and mRNA decay, providing the first functional evidence for the involvement of tRFs in Klotho-associated cognitive decline. Finally, we developed a novel cell type-specific short RNA dataset from live human brain tissues and used it to demonstrate that microRNAs changed by Klotho KO represent both neuron- and microglia-specific responses.

## Results

### Klotho knock-out induces significant perturbations in mRNA and long non-coding RNA profiles

Klotho deficiency perturbed 10% (1,059 out of 10,334) of all of the expressed polyadenylated RNA transcripts in Klotho KO murine brains compared to WT (padj < 0.05; Fig. 1a,b, Supplementary Data 1). Klotho mRNA itself was predictably downregulated in the brains of KO mice (padj«0.001; Supplementary Fig. 1). Other differentially expressed (DE) transcripts showed smaller albeit significant changes, possibly representing the averaged expression differences between diverse cell types and brain regions in the analyzed samples. Briefly, Klotho deficiency led to impaired functioning of multiple cellular compartments, for example, by modulating filamin B (*Flnb*), which is involved in cytoskeleton formation of developing neurons[16]; *Neat1*, a biomarker of nuclear paraspeckles[17]; *Slc6a6*, involved in synaptic GABA uptake[18]; and *Gatm*, a mitochondrial enzyme participating in creatinine synthesis[19]. Together, those differences clearly distinguished Klotho-knockout from wild-type samples in principal component analysis (PCA) (Fig. 1c).

Using the Gene Ontology (GO) resource[20] we identified the biological pathways of the differentially expressed (DE) genes. Klotho KO reduced protein folding and phosphorylation processes, indicating disrupted protein metabolism and exacerbated protein aggregation (Fig. 1d,e). Myelination and neuron projection development were also decreased in Klotho KO, while genes involved in cytoskeleton organization, transport, and dendritic spine morphogenesis were elevated, reflecting general morphological changes in neuronal structure (Fig. 1d,e). Considering that the analyzed bulk RNA samples represent mixtures of multiple brain cell types, we further deconvoluted the bulk RNA profiles using a single-cell RNA-seq mouse brain atlas[21]. An AutoGeneS package[22]-based cellular heterogeneity search indicated that Klotho KO caused a signaling decline in all glial cell types (Fig. 1f).

Out of 10,334 expressed polyadenylated RNA transcripts, 70 belonged to the long non-coding RNA (lncRNA) family (Supplementary Data 1). The change in their levels caused by Klotho KO was sufficient to separate the samples by genotype using PCA analysis (Supplementary Fig. 2a). Furthermore, the 14 differentially expressed lncRNAs included *Neat1, Miat* and *Meg3*, all upregulated in Klotho KO and involved in neurogenesis, neuron apoptosis regulation and neural inflammation[23–25] (padj « 0.05; Supplementary Fig. 2b).

The transcriptional perturbation caused by Klotho KO further mimicked changes seen in AD-mouse models. Meta-analysis of the AD-associated transcriptome[26] (Materials and Methods) revealed that the DE transcripts identified in the Klotho KO profiles highly overlapped with the mouse model AD-altered transcripts (Fig. 1g). Briefly, 811 out of the 1059 DE genes in Klotho KO ( ~ 77%) were altered in at least one of the mouse models, with 454 modulated in models with mutated amyloid precursor protein (APP; 43%), 104 – in models with mutated microtubule-associated protein tau (MAPT; 10%), and 253 in both (24%). The overlapping genes included known AD markers which changed in more than 7 different studies[26], including the lipid metabolism genes *Abca1* and *Apoe*[27]; complement immune system members *C4b, Ctsd, C1qb* and *C1qa*[28], and the astrocytic markers *Gfap* and *Aqp4*[29] (Supplementary Data 1). Thus, the transcriptional signature of murine

Klotho KO shared basic features with the transcriptomic profiles associated with diverse mouse AD models.

Notably, the observed links of altered RNA levels between Klotho KO and the AD-associated pathology are insufficient to suggest similarity between these phenotypes, since the levels of protein and their coding transcripts are not fully correlated[30]. Hence, we next evaluated the degree to which the mRNA changes induced by Klotho knockout are mirrored in the AD-associated proteome. Specifically, we calculated the proportion of the protein-coding transcripts, which were DE in Klotho KO and featured altered protein levels in a proteomic dataset from the mouse 5XFAD model[31] and in the compilation of proteomic datasets from human samples[32] (Methods). We found that 29% of the Klotho-DE transcripts encoded a protein that changed in a mouse 5XFAD model, 12% encoded a protein that changed in human AD proteomic studies, and 5.5% encoded a protein that changed in both (Fig. 1h; Supplementary Data 1). Notably, the DE mRNAs encoding for proteins with altered levels in human AD studies were statistically enriched in postsynaptic cytoskeleton organization and in tau protein binding, exposing the major common molecular cascades shared between the AD and the Klotho KO phenotypes (Fig. 1i).

### Klotho knock-out perturbs brain microRNA (miR) signatures

Apart from mRNAs, Klotho KO altered the levels of 27 miRs (Fig. 2a,b; Supplementary Data 2), whose signatures separated the samples on the PCA plane based on their genotype (Fig. 2c). Moreover, unsupervised clustering based on the profiles of the DE miRs reliably separated the samples to genotype-specific groups (Fig. 2d). Altered miRs include mmu-miR-212-5p, linked to longevity via regulating *Sirt1* and cooperating with the AD-depleted acetylcholinesterase-targeting mmu-miR-132-5p[33] (Fig. 2d). Identifying DE miRs targets using miRNet[34] revealed the experimentally validated targets whose levels were repressed by the corresponding miRs in similar experimental systems[35] (Materials and Methods; Supplementary Data 3). Further calculation of Pearson correlation coefficients between every pair of a DE miR and its DE-validated mRNA target enabled comparing the correlation coefficients between the miR – target pairs and miR – non-target pairs[36]. Intriguingly, miR-target profiles were more negatively correlated than miR- non-target pairs (Fig. 2e), demonstrating that some of the vast mRNA perturbations following Klotho KO might be explained by miR regulation. Using the Gene Ontology resource, we have further identified DE miR target genes involved in major neuronal pathways, such as axonogenesis, dendritic spine development, and regulation of synaptic plasticity (Fig. 2f). Further supporting active miR involvement in the Klotho-KO-activated downstream processes, we found Klotho KO -elevated levels of the microprocessor-associated protein DDX5, which is actively involved in miR processing[37], and is targeted by four different downregulated miRs (Supplementary Data 3).

To determine out if Klotho DE miRs are also altered in aging or AD pathology, we analyzed short RNA-seq data from AD mouse models, including young and aged (4 and 10 months) wildtype and transgenic mice carrying either *APP* (APP^swe/PS1^L166P) or *TAU* (THY-Tau22) mutations[38]. For each strain, we compared miR profiles from wildtype and age-matched transgenic mice, and between ages within the same genotype (Supplementary Data 4). This identified Klotho-associated miRs whose levels were affected by age (e. g. miR-488-3p; miR-323-3p), by both age and *APP* pathology (miR-150-5p, miR-27a-3p), by *APP* pathology alone (miR-434-3p, miR-129-5p), by *TAU* pathology alone (miR-7b-5p), or by both *APP* and *TAU* pathologies (miR-433-5p; Fig. 2g,h), together reflecting altered regulation of cognition.

Next, we sought putative human links between DE miRs, cognition, and AD neuropathology. Intriguingly, two miRs that were DE in Klotho KO, miR-129-5p, which is associated with the Fragile X syndrome[39] and the muscarinic receptors-targeting miR-335-5p[40], were reduced in the CSF of female, but not male AD patients[41] (Fig. 2i). Considering our recent findings of cognition-related female-specific AD declines in tRF regulators of cholinergic mRNAs[26], this suggests links between Klotho-KO-induced transcriptomic perturbations and sex differences in dementia. Indeed, both

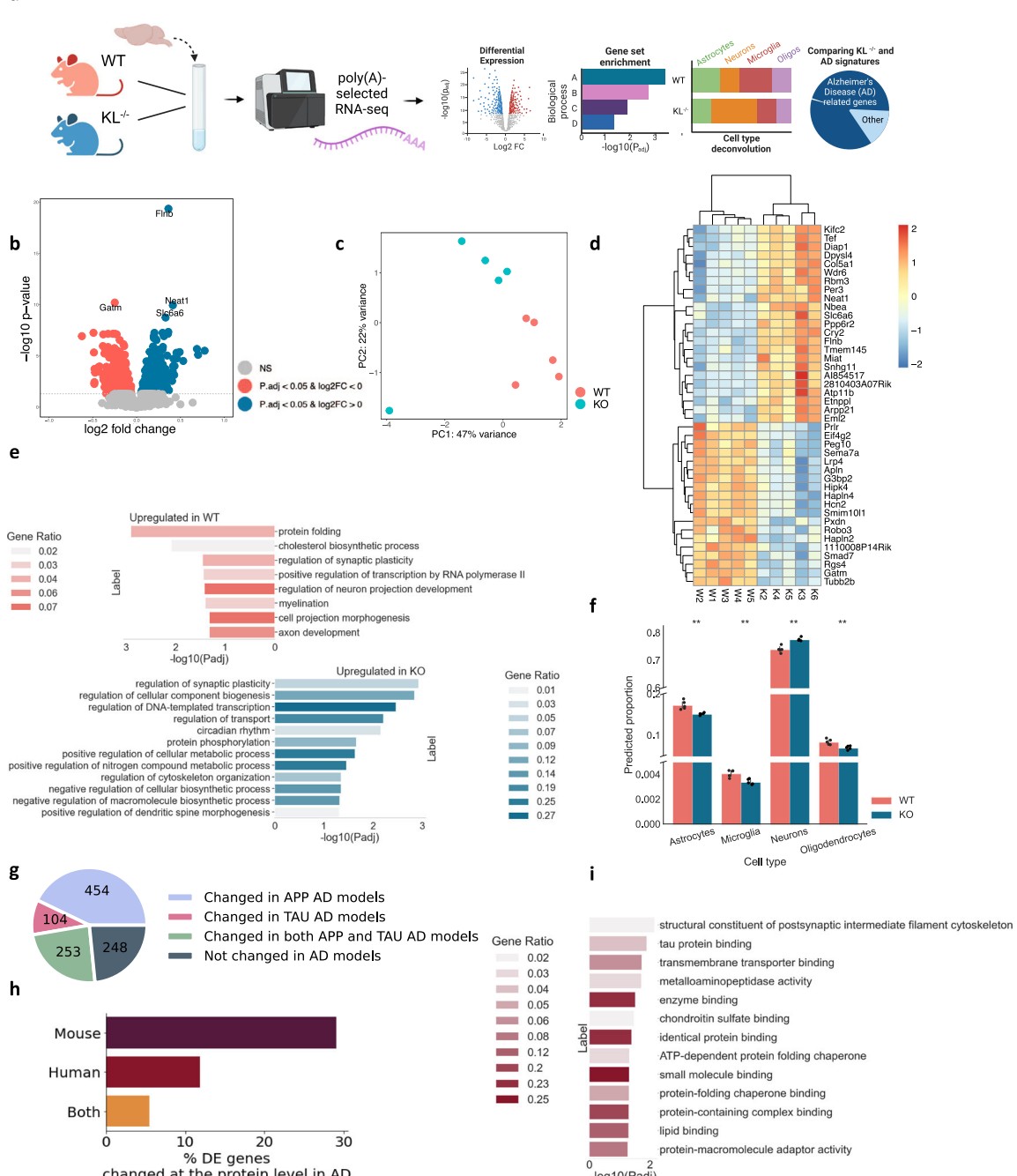

**Fig. 1 | Klotho KO induces numerous long RNA transcriptome changes.**
**a** Graphical representation of the analysis shown in Fig. 1. Created with BioRender.
com. **b** Volcano plot of polyadenylated RNAs DE in murine Klotho-knockout
($n = 5$) compared to wildtype ($n = 5$) brains; significantly upregulated transcripts in
blue, significantly downregulated transcripts in pink; $p_{adj} < =0.05$. **c** PCA of poly-
adenylated RNA reads colored by genotype. **d** Heatmap showing normalized and
centered counts of the most significantly altered transcripts (padj <= 5e-6; colormap
- z-score of normalized count levels, centered to the mean). **e** GO annotation of
biological processes enriched in up- and down-regulated transcripts (blue and pink,

respectively) under Klotho-knockout. **f** AutoGeneS prediction of the cell type pro-
portions in WT and KO profiles based on the scRNA-seq atlas[21]; **: pval < 0.01, ns:
non-significant. **g** Pie chart proportions of DE genes in AD studies[26]. **h** Bar plot
showing the percent of DE mRNAs encoding proteins that are altered in a mouse
model AD study[31], in a compilation of human AD proteomic studies[32] and in both. **i**.
GO annotation of molecular functions enriched in protein-coding transcripts whose
associated proteins were altered in a compilation of human AD proteomic studies[32].
The error bands denote 95% confidence intervals.

miR-129-5p and miR-335-5p are also altered in the nucleus accumbens of
female AD patients, compared to controls with no cognitive impairment[15]
(Fig. 2j; Supplementary Data 5), and with the same direction of change in
human AD and murine Klotho KO brains. These down-regulated tran-
scripts and processes hence emerge as co-related to the cognitive impair-
ments shared by Klotho KO and AD.

## Klotho knockout alters the levels of short tRNA fragments (tRFs)
Apart from miRs, we also identified Klotho-associated expression changes
in another short non-coding RNA family – tRFs[11] (Fig. 3a). Six nuclear
genome-originated tRFs were altered, three upregulated and three down-
regulated (Fig. 3b; Supplementary Data 6). Strikingly, all three upregulated
tRFs were 3' halves originated from Asparagine-GTT tRNA, representing

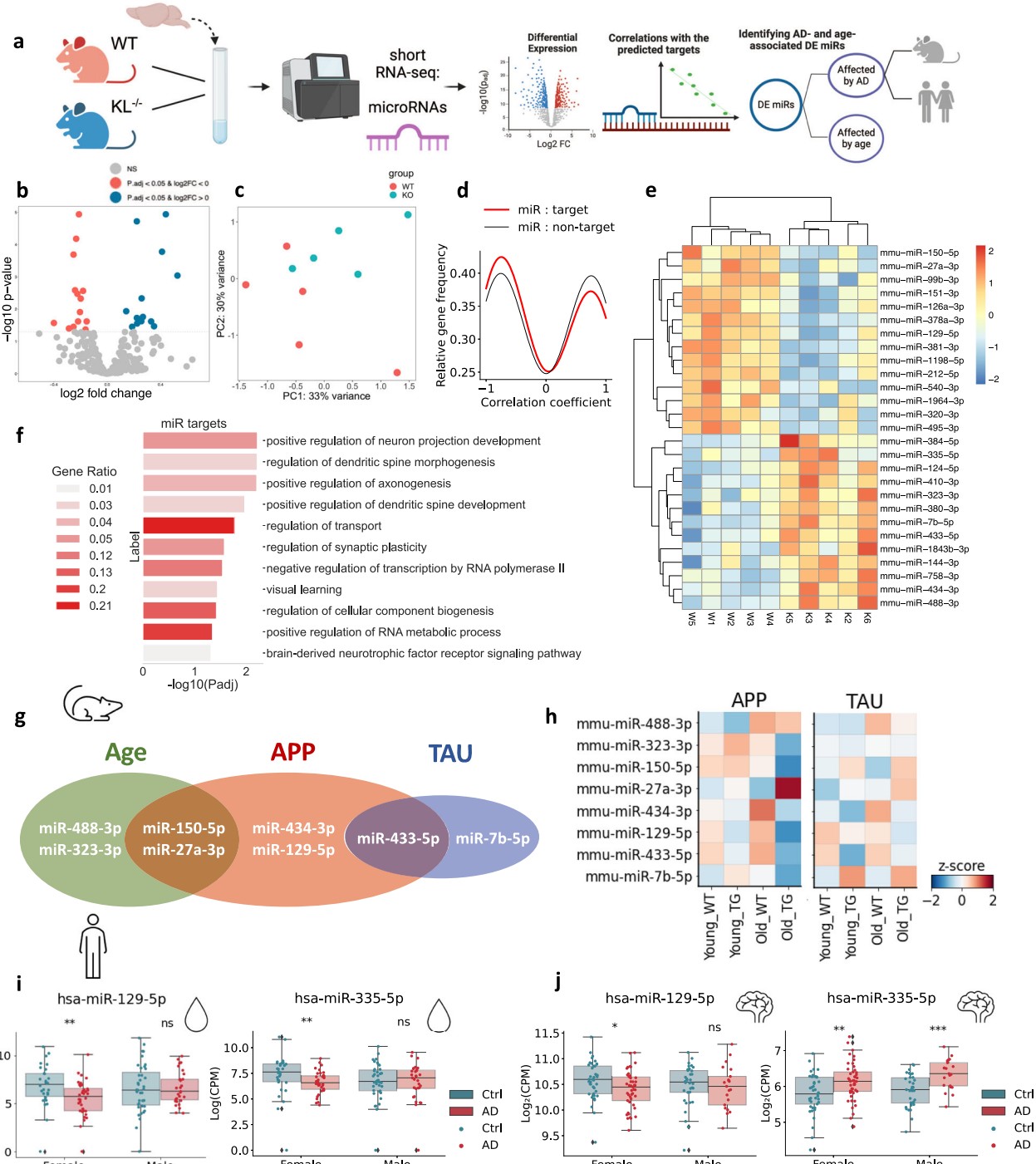

**Fig. 2 | Klotho depletion perturbs miR signatures, resembling AD and aging-associated changes. a** Graphical representation of the analysis shown in Fig. 2. Created with BioRender.com. **b** Volcano plot of miRs DE in Klotho-knockout ($n = 5$) compared to wildtype ($n = 5$) mice, significantly upregulated transcripts in blue, significantly downregulated transcripts in pink, $p_{adj} < =0.05$. **c** PCA based on miR counts and colored by genotype. **d** Gaussian kernel density estimates of Pearson correlation coefficients between klotho-affected miRs and their targets (red), and klotho-miRs and DE mRNAs which are not their targets (black). X axis: Pearson correlation coefficient. Y axis: relative frequency of miR-mRNA pairs with the corresponding coefficient. **e** Heatmap showing normalized and centered counts of DE miRs (colormap: z-score of the normalized counts, centered to the mean). **f** Gene

Ontology annotation of miR targets from enriched biological processes. **g** Venn diagram of DE miRs that co-change in the murine hippocampus with age and/or in one of the AD pathology models[38] (APP - APP^swe/PS1^L166P; TAU - THY-Tau22). **h** Scaled heatmap of miR alterations from (G) across age and genotype conditions of both APP and TAU mouse models[38]. **i** Log2(CPM) counts of hsa-miR-129-5p and hsa-miR-335-5p in CSF from AD patients and cognitively healthy controls, divided by sex[41] (**: $p_{adj} < 0.01$). **j** Log2(CPM) counts of hsa-miR-129-5p and hsa-miR-335-5p in the nucleus accumbens of AD patients (cogdx=4), compared to cognitively healthy individuals (cogdx=1), divided by sex[15] (**: $p_{adj} < 0.01$). The error bands denote 95% confidence intervals.

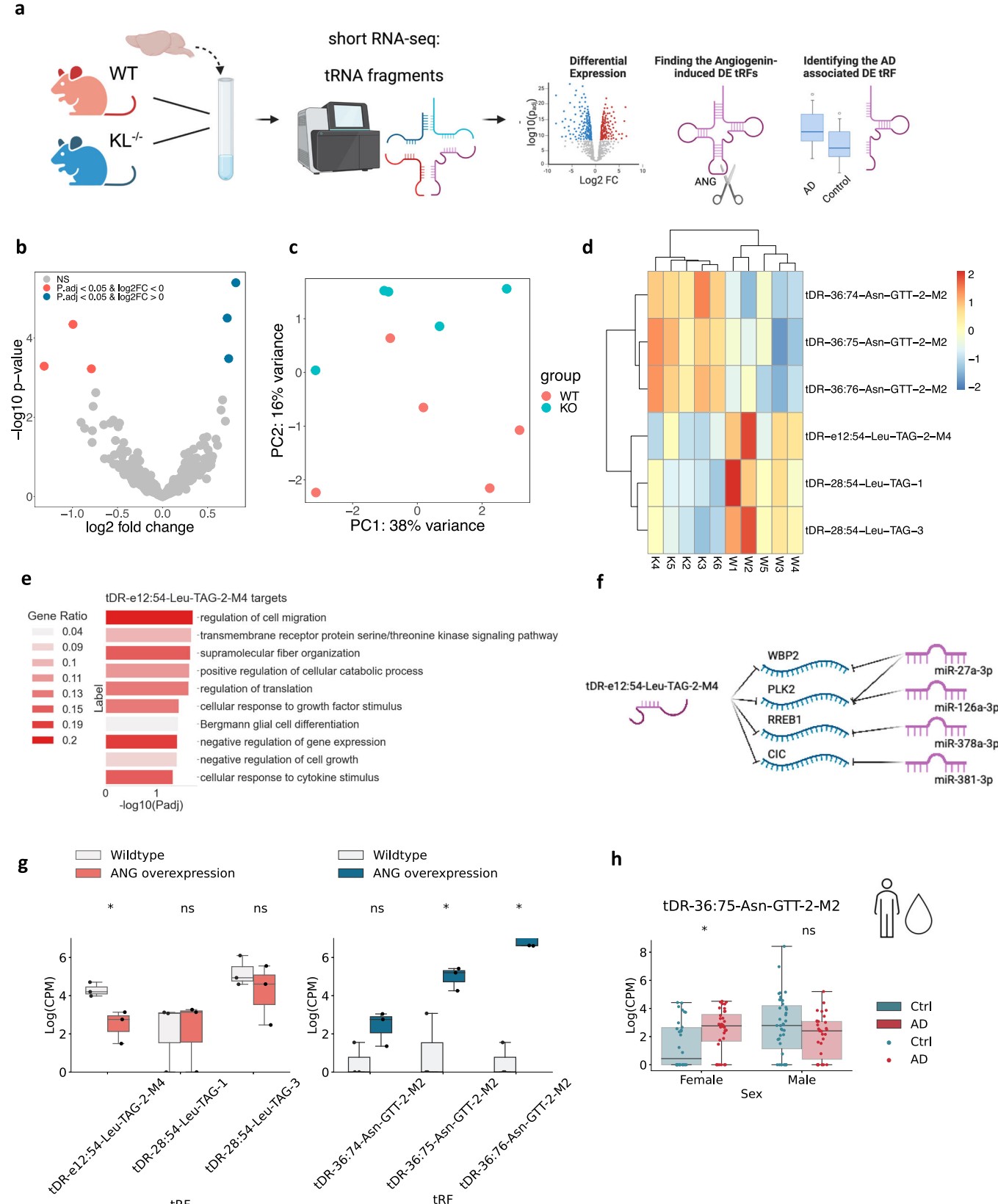

matching 3'-end sequences with single nucleotide differences, whereas the three downregulated tRFs were Leucine-TAG tRNA-originated i-tRFs (Fig. 3b; Supplementary Data 6). These subtle differences allowed PCA to separate the samples (Fig. 3c), and unsupervised hierarchical clustering based on the DE tRFs recognized the knockout and wildtype genotypes as two distinct clusters (Fig. 3d).

Of all the DE tRFs, only tDR-e12:54-Leu-TAG-2-M4, downregulated in Klotho KO, was short enough to predict miR-like activity[14]. The tRFTar

**Fig. 3 | Klotho knockout alters tRF levels sex-dependently. a** Graphical representation of the analysis shown in Fig. 3. Created with BioRender.com. **b** Volcano plot of DE tRFs in murine Klotho-knockout (*n* = 5) compared to wildtype (*n* = 5) brains; significantly upregulated transcripts in blue, significantly downregulated transcripts in pink; $p_{adj}$ < =0.05. **c** PCA of tRF counts colored by genotype. **d** Heatmap showing normalized and centered counts of altered tRFs (colormap: z-score of the normalized counts, centered to the mean). **e** Gene Ontology annotation of biological processes enriched in tDR-36:75-Asn-GTT-2-M2 targets, and negatively correlated with this tRF in Klotho KO data. **f** Schematically illustrated common targets of tDR-36:75-Asn-GTT-2-M2 and miRs. **g** Log$_2$(CPM) counts of altered tRFs in HEK293 cells under angiogenin overexpression compared to wildtype; left panel, tRFs decreased, right panel tRFs increased under Klotho KO;[43].*: $p_{adj}$ < 0.1. **h** Log (CPM) counts of tDR-36:75-Asn-GTT-2-M2 in CSF of AD patients and cognitively healthy controls, divided by sex[41],*: $p_{adj}$ < 0.05. The error bands denote 95% confidence intervals.

tool[42] identified 81 predicted mRNA targets that were negatively correlated with tDR-e12:54-Leu-TAG-2-M4 (Supplementary Data 7). Notably, these tRF targets were enriched in biological processes distinct from those enriched by the Klotho-altered miR targets, and included cell migration, intracellular signaling, and translation regulation pathways (Fig. 3e). Moreover, tDR-e12:54-Leu-TAG-2-M4 shared 4 targets with downregulated miRs (Fig. 3f), possibly reflecting cooperation of these short noncoding RNA families in knockout-induced transcriptional changes.

The transcriptomic profiles further revealed slightly upregulated mRNA levels of the nuclease angiogenin, which is linked to stress-related tRF production (Supplementary Fig. 3b)[43]. To determine whether tRFs elevated by Klotho knockout were stress-induced products of angiogenin[43,44], we sought parallel tRF changes in relevant model systems. Notably, all Klotho KO upregulated tRFs were found in angiogenin-overexpressing HEK293 cells, and one of the downregulated tRFs, tDR-e12:54-Leu-TAG-2-M4, decreases in U2OS cells with angiogenin overexpression (Fig. 3g) and increases in angiogenin knockout[43] (Supplementary Fig. 3a). Thus, the tRF signature following Klotho knockout might represent a stress-induced and angiogenin-dependent response.

Comparing the identified tRF changes under Klotho knockout to those of AD brains, identified tDR-36:75-Asn-GTT-2-M2 as both upregulated in Klotho knockout and in the CSF of female AD patients compared to cognitively unimpaired controls[41] (Fig. 3h). Mimicking the sex-specific miR changes and decline of tRFs in the nucleus accumbens from AD female brains[15], tDR-36:75-Asn-GTT-2-M2 levels remained unchanged in male patient brains. In summary, both Klotho knockout-induced brain miR and tRF changes correspond to their parallel changes in AD pathology.

## tDR-36:74-Asn-GTT-2-M2 interacts with proteins involved in RNA splicing and degradation

We further aimed to establish a functional role of the tRFs upregulated in Klotho KO, which are of particular interest for several reasons. First, these tRFs only differ in 1-2 nucleotides at the 3' end, essentially representing the same sequence, which constitutes a highly specific response and may suggest a functional impact for the shared sequence. Second, these tRFs belong to the family of stress-related tRNA halves[14] and might be involved in angiogenin-induced stress response, caused by Klotho KO (Fig. 3g; Supplementary Fig. 3a). Finally, tDR-36:75-Asn-GTT-2-M2 is elevated in the CSF of female AD patients (Fig. 3h), providing additional evidence for functional role in neurodegeneration and cognitive decline.

Over the past few years, tRNA halves are continuously reported to modulate crucial intracellular pathways, such as translation and apoptosis, by direct interaction with proteins[14,45,46]. Therefore, we sought proteins interacting with the tRFs of interest. To this aim, we established a pull-down assay in which we incubated a lysate of SH-SY5Y human neuroblastoma cells with a biotinylated sequence of the shortest tRF upregulated in Klotho KO, tDR-36:74-Asn-GTT-2-M2, representing the shared sequence between the three upregulated fragments. Importantly, this tRF is conserved between the human and mouse genomes, which enabled us to perform the experiment using a human cell line. The proteins bound to the sequence were then isolated using streptavidin-coated magnetic beads and identified by mass spectrometry (MS) (Fig. 4a). Close to 100 proteins were found to be significantly enriched in the tDR-36:74-Asn-GTT-2-M2 samples, compared to the samples treated with a negative control (NC) oligo, allowing for a clear separation on the PCA plane (Fig. 4b; Supplementary Data 8). We further focused our analysis on the 13 proteins which were most enriched in tDR-36:74-Asn-GTT-2-M2 compared to NC samples, using a fold change (FC) cutoff of log$_2$(FC)≥4, to minimize analysis of proteins not markedly changed between the tRF and the NC (Fig. 4c,d; Materials and Methods).

Intriguingly, these 13 proteins represent a highly interconnected molecular network associated with RNA processing. The most enriched biological process among these proteins is regulation of RNA splicing (Fig. 4e). GRSF1, HNRNPF, HNRNPH3, SNRNP70, NCL and HNRNPH1 are annotated as parts of the spliceosome complex by the GO resource[20], along with SART3 and SNRPA that are reported to be involved in RNA splicing as well[47,48]. Thus, tDR-36:74-Asn-GTT-2-M2 might be a potential regulator of alternative splicing, which represents a hallmark of aging and neurodegeneration[49–51].

Another highly relevant pathway, represented by PUS7 and TRUB1, is RNA pseudouridylation[52,53] (Fig. 4e). G-rich pseudouridylated tRNA-fragments were shown to control translation, eventually determining the fate of the developing hematopoietic stem cells[54] and predicting leukaemic progression[55]. Although this effect has been demonstrated for 5' halves[54,55], our current results indicate that 3' halves, such as tDR-36:74-Asn-GTT-2-M2, might be modified as well, possibly exerting a parallel downstream function. Finally, SKIV2L and TTC37 represent components of the SKI complex, which is involved in 3' to 5' mRNA decay[56].

Notably, PUS7, CNBP, GRSF1, and HNRNPF specifically bind to G-rich oligonucleotide sequences or G-quadruplexes[54,55,57–59], with CNBP promoting their formation[57]. This is particularly relevant, since G-rich tRNA halves were demonstrated to form G-quadruplexes, thereby inhibiting translation and promoting neuroprotection via the formation of stress granules[60]. Guanine is indeed the most frequent base in the tDR-36:74-Asn-GTT-2-M2 sequence, consisting of 33% G, 28% C, 23% A, and 16% T (Materials and Methods). Therefore, we hypothesize that tDR-36:74-Asn-GTT-2-M2 and its analogous fragments, highly expressed under Klotho KO, have the potential to form G-quadruplexes. This secondary structure may facilitate their molecular roles, thus affecting splicing, mRNA decay or translation[60].

## Klotho-miR profiles reveal neuronal and microglial signatures

Deconvoluting the bulk mRNA signal to cell type-specific signatures showed that Klotho KO led to reduced numbers of glial transcripts (Fig. 1f). However, deconvoluting the affected miRs, which are highly cell type-specific[61], was challenging due to the lack of reference datasets with cell type resolution. Therefore, we developed novel neuron- and microglia-specific short RNA-seq datasets from live human brain samples resected during surgeries for the removal of non-infiltrative brain tumors. To assess the non-tumor origin of the cortical tissues, we excluded patients with infiltrative brain pathologies such as meningitis, glioma, or diffuse cerebral inflammation, and collected the tissues as far from the tumor as possible. We then validated that the acquired miR profiles differ from specimens of meningioma[62], but rather resemble postmortem bulk miRNA profiles from various human brain regions[63] (Supplementary Fig. 4a-c).

We collected 16 live human brain samples (6 females, 10 males; 42-75 years old) and recorded a wide range of patients' additional medical parameters that could be relevant for future analyses (Supplementary Data 9). Following our inhouse NuNeX protocol[64], the collected samples were exposed to a brief formalin fixation and separated into single nuclei suspensions, which retained some associated cytoplasm. Staining with neuron-

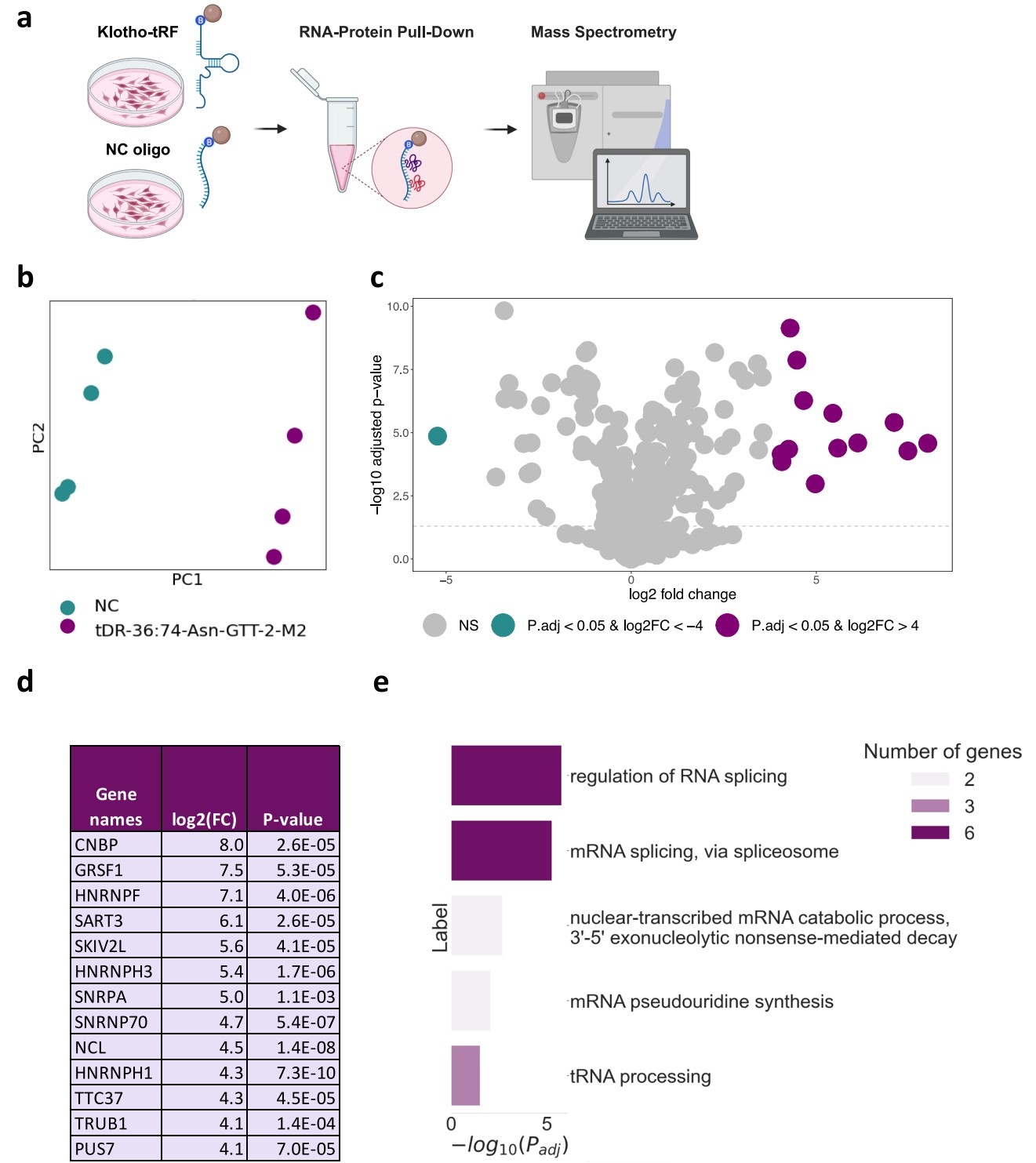

**Fig. 4 | Pull-down assay revealing that tDR-36:74-Asn-GTT-2-M2 binds multiple proteins involved in splicing, mRNA decay and pseudouridylation. a** Graphical representation of the pull-down experiment: SH-SY5Y cells were plated, harvested after four days, and incubated with magnetic streptavidin beads to which biotinylated tDR-36:74-Asn-GTT-2-M2 (Klotho-tRF) or biotinylated negative control (NC) RNA-oligonucleotides were bound. After wash steps the beads were submitted to mass spectrometry and the proteins bound to each oligonucleotide were identified. Created with BioRender.com. **b** PCA showing separation of the proteins bound to the tDR-36:74-Asn-GTT-2-M2 (purple) vs. those bound to the NC oligo (green),

4 samples per group. **c** Volcano plot showing the differentially expressed proteins in biotinylated tDR-36:74-Asn-GTT-2-M2 ($n = 4$) vs biotinylated NC pulldown samples ($n = 4$). Proteins significantly upregulated with log2FC > 4 are colored in purple, proteins sinificantly downregulated with log2FC > =-4 are colored in green. The statistical analysis in was performed using the Perseus package. **d** List of the 13 proteins enriched by log2FC > =4 (16-fold) in tDR-36:74-Asn-GTT-2-M2 vs. NC samples. **e** Gene Ontology annotation of the biological processes enriched within the differentially expressed proteins in panel D.

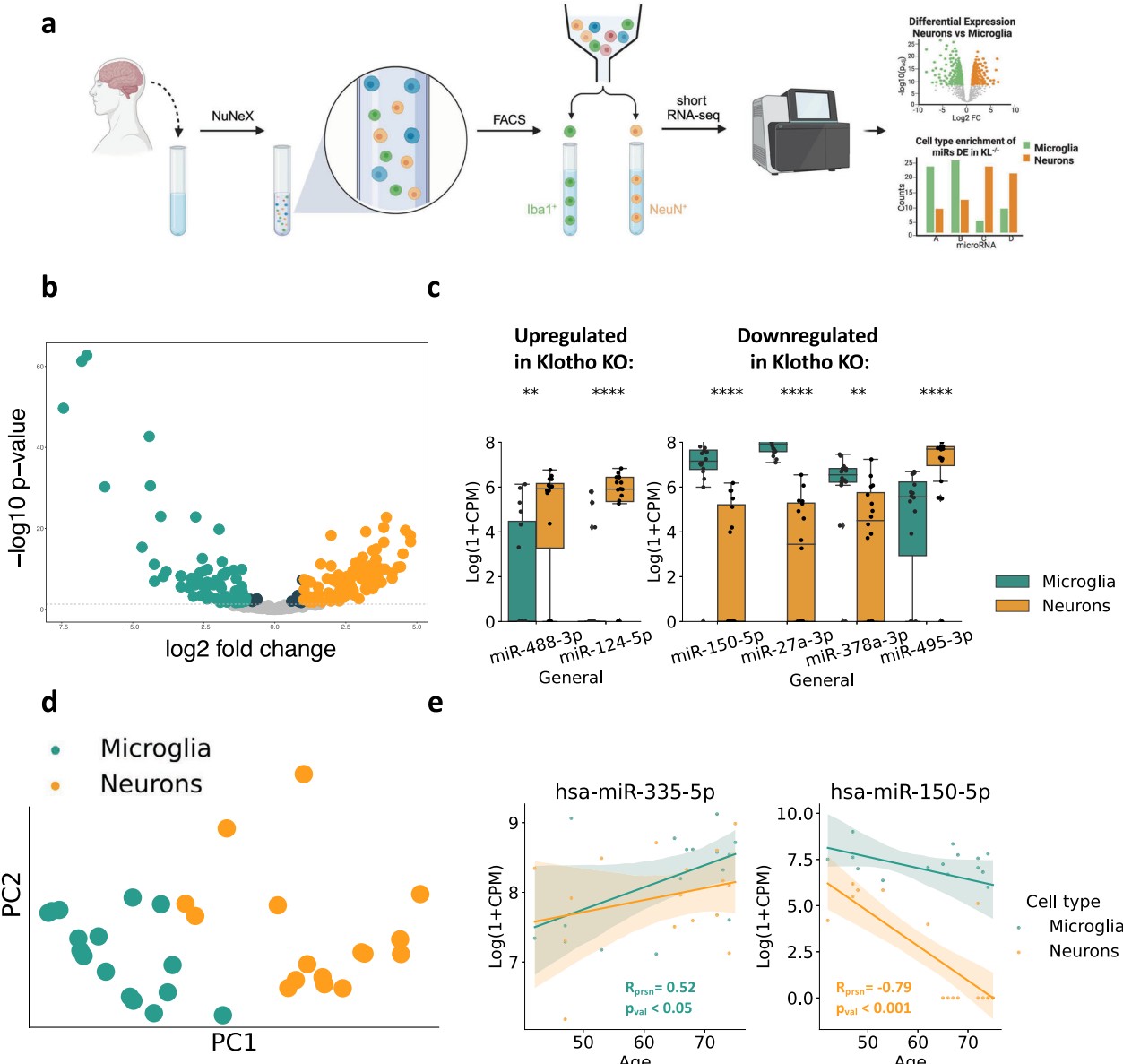

**Fig. 5 | Klotho KO affects miR levels in both neurons and microglia. a** Graphical representation of the pipeline used to produce bulk cell type-specific short transcriptome datasets of live brain cells. Using the NuNeX protocol[64], samples extracted during neurosurgeries were broken down to single nuclei surrounded by thin cytoplasm layers. Nuclei stained with microglia- (Iba1) and neuron- (NeuN) specific fluorescent markers were FACS-sorted and short RNA-sequenced. Created with BioRender.com. **b** Volcano plot showing altered level miRs enriched in neurons ($n = 16$) in orange or microglia ($n = 16$) in green. **c** 6 miRS DE in Klotho KO were also enriched in one of the cell types. Left panel: miRs upregulated in Klotho KO; Right panel: miRs downregulated in Klotho KO, **: $p_{adj} < =0.01$; ****: $p_{adj} < =0.0001$. **d** PCA of the cell type-specific small transcripts (7 miRs DE in Klotho KO, colored by cell type). **e** Linear correlation of donor age with $\log_2(CPM)$ count levels of hsa-miR-335-5p (significantly correlated in microglia) and hsa-miR-150-5p (significantly correlated in neurons), colored by cell type as in panel D. The error bands denote 95% confidence intervals.

specific (NeuN) and microglia-specific (Iba1) fluorescent markers followed by fluorescence-activated cell sorting (FACS; Supplementary Fig. 5a–e) yielded 32 neuron- and microglia-specific populations and their short RNA-seq profiles (Fig. 5a; Supplementary Datas 10, 11). Since Iba1 is also expressed by macrophages that infiltrate the brain[65], we validated the microglial identity of the sorted Iba1+ populations using qPCR measurement of specific microglial RNA markers, such as *P2y12*, *Sall1*, and *Tmem119* (Supplementary Fig. 6).

Intriguingly, our novel short RNA-seq datasets revealed highly distinct miR profiles between live human brain neurons and microglia (Fig. 5b). Using the DESEQ2 package[66], we found that 57% of the expressed miRs (173 of 303) were enriched in one of these cell types (Supplementary Data 10). Among them, 69 and 91 were upregulated in microglia or neurons with a

log-fold change higher than 1, yielding highly valuable resources for deconvolution of bulk brain-derived short RNA profiles. Furthermore, the miRBase microRNA annotation database features 1830 mature human miRs and 1650 mature murine miRs of mouse origin, 493 of which are conserved and represent human-murine orthologs, which enables usage of our cell-type resolved human dataset for deconvoluting bulk data from murine tissues as well[67].

Of the 27 DE miRs in murine Klotho KO, 7 were found in our newly produced cell type-specific human dataset, with 6 of those selectively enriched in one of those cell types (Fig. 5c; Supplementary Data 10). Parallel to the decline in glial mRNA transcripts under Klotho KO, 3 out of 4 miRs decreased in Klotho KO were enriched microglia from live human brain samples (Fig. 5c). In contrast, miRs upregulated in Klotho KO were enriched

in human neurons from live human brain samples, mimicking the elevated neuron-specific mRNA transcripts in our mRNA-based deconvolution. Further, the cell type-dependent expression of the 7 DE miRs in Klotho KO sufficed to separate microglial and neuronal samples, showing a strong cell type-specific signal in the Klotho KO miR profiles (Fig. 5d) with shared cell type specificity of the DE miRs seen between the sexes (Supplementary Fig. 7a-c).

Notably, two DE miRs in Klotho KO showed cell type-dependent age correlation (Fig. 5e) The cholinergic miR-335-5p, which targets the muscarinic receptors and has been proposed to limit the cholinergic blockade of inflammation[40,68], was upregulated with age in human brain microglia, murine Klotho KO and AD patients' CSF and nucleus accumbens (Fig. 2i,j). In contrast, miR-150-5p, which targets the autism-associated Foxp1[69] transcript, declined with age in live human neurons, Klotho KO and by both age and APP pathology in AD mice models (Fig. 2g, h). Thus, our findings further strengthened the notion of miR-335-5p and miR-150-5p contributions to aging-related processes.

Aligning our neuron- and microglia-specific short RNA datasets to a tRF database (Materials and Methods) produced a novel resource of cell type-specific tRFs from live human brains. Unlike miRs, there were many more tRFs elevated in neurons than in microglia (Supplementary Fig. 8a). While none of the tRFs DE in Klotho KO were expressed in human neurons or microglia, other fragments originating from the same tRNAs exhibited a trend of elevation in neurons as compared to microglia (Supplementary Fig. 8b), possibly indicating that tRF signaling is generally more potent in neurons than in microglia. The neuron-specific tRF dataset is openly available (Supplementary Data 11).

## Discussion

Klotho is an anti-aging gene with promising therapeutic potential[70]. To seek mechanisms by which Klotho exerts its CNS effects, we compared the RNA profiles of murine Klotho-deficient brains with those of healthy controls. This revealed a major Klotho knockout-induced transcriptional dysregulation affecting multiple cellular compartments and brain cell types. Importantly, many Klotho KO-altered mRNA transcripts and their protein products were also changed in brain aging and AD. Remarkably, short non-coding RNA transcripts, affected by Klotho KO, also exhibit parallel changes in aging and AD, reflecting an orchestrated transcriptional response shared by Klotho KO and aging-related cognitive decline.

While the potential involvement of tRFs in major molecular pathways has been established, their role in neurodegenerative diseases and cognitive decline is incompletely understood. We provide the first experimental evidence of the functional involvement of tRFs in Klotho-related cognitive decline. Specifically, Klotho KO elevated the expression of tDR-36:74-Asn-GTT-2-M2, which we show to bind numerous spliceosome components and major parts of the SKI complex involved in mRNA decay. In addition, our results hint that this tRF may be subjected to pseudouridylation and thereby control translation. Further experimental validation is required to establish the functional impact of the interaction between tDR-36:74-Asn-GTT-2-M2 and the reported pathways, as well as the role of its pseudouridylation[71].

Lastly, our study substantially contributes to the understanding of the cell type specificity of brain-derived short non-coding RNAs. The deconvolution analysis of Klotho KO-altered mRNA profiles using a single-cell RNA-seq atlas of the murine brain[21,22] showed that Klotho knockout causes a slight decline in glial mRNA transcripts. By generating a novel cell type-resolved short RNA dataset from live human brain tissues, we managed to support this finding at the level of microRNA, revealing that Klotho KO led to reduced levels of microglia-enriched miRs while elevating the levels of neuronal miRs, thus providing a link to potential mechanisms of action explaining at least part of its aging-related roles. Our novel cell-type resolved dataset, exhibiting neuron- and microglia-specific expression profiles of miRs and tRFs, provides a key resource for aging researchers with a focus on Klotho-associated brain processes.

This study has several potential limitations. First, our discovery of overlapping molecular signatures between Klotho KO, human brain aging, and AD is correlative and requires experimental validation. Furthermore, our novel cell type-specific short RNA-seq dataset from live human brain tissues is prone to bias by the pathology effect. We have shown that the extracted profiles are closer to those of healthy postmortem brains than to meningioma samples (Supplementary Fig. 3a,b), similar to multiple previous studies on live human brain tissue, which suffered from the same limitation[72-74]. However, we cannot fully rule out the possibility that the extracted neuronal and microglial profiles are affected by the tumor. Finally, the NuNeX protocol yields mainly nuclear-enriched RNA profiles, introducing a bias into the deconvolution analysis and possibly omitting the signal from transcripts abundant in other cellular compartments such as synapses.

In summary, while the major phenotypic effect of Klotho deficiency on premature aging and dementia has already been established, our results contribute to the study of Klotho-associated aging processes by thoroughly describing the transcriptomic changes occurring in Klotho knock-out and providing the first notes referring to its impact on tRFs-affected brain proteins and their related pathways. Overall, the data presented here indicates that physiologically normal Klotho levels are necessary for white and gray matter development and maintenance throughout life. Since Klotho levels decrease with age, augmenting Klotho protein expression either by enhancing its endogenous expression or by delivering exogenous genetic material, could be beneficial in neurodegenerative and other age-related diseases. We believe that by exposing the transcriptional network perturbation caused by the deficiency of Klotho, our work will accelerate future drug discovery targeting aging-related processes.

## Methods
### RNA extraction from Klotho KO murine brains
Klotho knock-out ($KL^{-/-}$) mice were generated as described previously[75,76], by crossing $KL^{+/-}$ mice carrying an insert mutation at 5' promoter region of the KL gene. WT and $KL^{-/-}$ female mice, $n = 5$ in each group, were sacrificed at 6 weeks of age by cervical dislocation and the whole brain removed and snap-frozen in liquid nitrogen before storage at -80°C. Brains were homogenized in QIAzol (Qiagen, 217004, kit component) using a Polytron PT 3000 (Kinematica) and stored at -80°C. RNA was extracted from a 700 μL aliquot of each homogenate using the miRNeasy Mini Kit (Qiagen, 217004) according to the manufacturer's protocol. RNA concentration was determined (NanoDrop 2000, Thermo Scientific) and RIN was measured (Bioanalyzer 6000, Agilent), with all samples ranging from 8.3 to 8.9.

### RNA sequencing and alignment of Klotho KO murine brain-derived RNA
All samples underwent small RNA-sequencing and poly(A)-selected long RNA-sequencing. For small RNA, libraries were constructed from 1000 ng total RNA (NEBNext Multiplex Small RNA library prep set for Illumina, New England Biolabs, NEB-E7560S), and the small RNA fraction sequenced on the NextSeq 500 System (Illumina) at the Center for Genomic Technologies Facility, the Hebrew University of Jerusalem. For poly(A)-selected RNA, libraries were constructed from 1000 ng total RNA (KAPA Stranded mRNA-Seq Kit, Kapa Biosystems, KR0960–v5.17), and the RNA sequenced as above. Small RNA was aligned to miRBase using miRexpress[77] and to the tRNA transcriptome using the MINTmap[78]. Long RNA was aligned to the Mus Musculus reference genome (GRCm38) using STAR[79]. The nomenclature of tRFs was based on the recently published standardized ontology[80].

### Cell culture for tRF pull-down assay
The human-derived neuroblastoma line SH-SY5Y (ATCC, CRL-2266) was grown in standard conditions at 37°C and 5% CO2, in EMEM:F12 (Merck, M5650 and N4888, respectively). All media were supplemented with FCS (10% final concentration, Sartorius, 04-127), L-glutamine (2 mM final

concentration, Sartorius, 03-020) and Penicillin-Streptomycin-Amphotericin (100 units/mL, 0.10 mg/mL, 0.25 μg/mL, final concentrations, respectively, Sartorius, 03-033). Cells were mycoplasma-free (Mycoblue® Mycoplasma Detection Kit, Vazyme, D101).

### RNA-Protein pull-down assay

The Pierce™ Magnetic RNA-Protein Pull-Down Kit (Thermo Fisher Scientific, 20164) was used to identify proteins binding to tDR-36:74-Asn-GTT-2-M2. Cells were seeded in 100 ×15 mm plates (Thermo Fisher Scientific, 150350) at 10-12×10^6 cells/plate and harvested four days later in 500 μL lysis buffer/plate. Lysis buffer contained 25 mM Tris-HCl pH 7.4, 150 mM NaCl, 1% IGEPAL® CA-630, 1 mM EDTA pH 8.0, and 5% glycerol (identical to Pierce® IP Lysis Buffer, Thermo Fisher Scientific, 87787), and Protease Inhibitor Cocktail (1:100, Cell Signaling Technology, 5871). Plates were incubated on ice for 15 minutes, cells were scraped off, collected in Eppendorf test tubes, and incubated another 5 minutes on ice. After samples were clarified by high-speed centrifugation (10 minutes, 13,000 g, 4°C) and transferred to fresh test tubes, protein concentrations were determined using the Bradford assay (Sigma Aldrich, B6916). Standards and samples were prepared in diluted lysis buffer (1:1) to reduce IGEPAL® concentration to 0.5% which is compatible with the assay. Streptavidin magnetic beads (Dynabeads™ MyOne™ Streptavidin C1, Thermo Fisher Scientific, 65001), at 50 μL/sample, were pre-washed per kit instructions and 100 pmol/sample of 5'-biotinylated RNA oligonucleotide was bound to the beads per kit instructions (we did not use the 3' desthiobiotinylation option available in the kit). tDR-36:74-Asn-GTT-2-M2 sequence was as follows: UAA CCG AAA GGU UGG UGG UUC GAG CCC ACC CAG GGA CGC. The control sequence (NC5, IDT) was as follows: GCG ACU AUA CGC GCA AUA UG. In both cases the penultimate base and the one preceding it were protected by 2'-O-methylation. Beads were washed again and 120 μg cell lysate was added to each sample, with glycerol and Tween® 20 at final concentrations of 15% and 0.1%, respectively, per the initial conditions suggested in the kit. After rotation for 1 hour at 4°C beads were washed once in kit Wash Buffer and another three times in the same buffer lacking detergent. 40% of the bead pellet from each sample (representing 48 μg protein/sample) was stored at -80°C until submission for mass spectrometry.

### Mass spectrometry

On-bead digestion was performed as follows: packed beads were resuspended in buffer containing 8 M urea, 10 mM DTT, and 25 mM Tris-HCl pH 8.0, and incubated for 30 minutes. 55 mM iodoacetamide was then added and beads incubated for an additional 30 minutes. The urea was diluted in 8 volumes of 25 mM Tris-HCl pH 8.0 and 0.4 μg/sample trypsin added. Beads were incubated overnight at 37°C with gentle agitation, peptides were acidified in 0.38% formic acid, and desalted. MS was performed on a Q Exactive™ Plus mass spectrometer coupled on-line to a Dionex UltiMate™ 3000 system, with Xcalibur™ software for data acquisition (Thermo Fisher Scientific). Peptides were separated over an acetonitrile gradient (0 - 80%), flow rate 0.15-0.3 μl/min on a reverse phase 25-cm-long C18 column (75 μm ID, 2 μm, 100 Å, PepMap RSLC) for 120 minutes. Survey scans (380–2,000 m/z, target value 3E6 charges, maximum ion injection times 50 ms) were acquired and followed by higher energy collisional dissociation-based fragmentation. A resolution of 70,000 was used for survey scans and up to 15 dynamically chosen most abundant precursor ions with "peptide preferred" profile were fragmented (isolation window 1.8 m/z). The MS/MS scans were acquired at a resolution of 17,500 (target value 1E5 charges, maximum ion injection times 120 ms). Dynamic exclusion was 60 sec. To avoid carryover the column was washed with 80% acetonitrile and 0.1% formic acid for 25 min between samples. Data was processed using the MaxQuant computational platform. Peak lists were searched against the human reference proteome from UniProt (UP000005640) and allowed up to two mis-cleavages. Peptides with a length of at least seven amino acids were analyzed and the required FDR was set to 1% at the peptide and protein level. Relative protein quantification was determined using the label-free quantification (LFQ) algorithm. The Perseus statistical package with default software parameters was used for statistical computations. All analysis was performed at the Stein Family mass spectrometry center in the Silberman Institute of Life Sciences, The Hebrew University of Jerusalem.

### Sampling of live human brain tissue

The study, as well as acquisition of cortical brain samples, was approved by the institutional review board (IRB protocol RMB-0713-19) and by the national ethical review board. All patients signed informed consent forms. The biopsies were extracted in operations for resection of non-neural pathologies, such as meningiomas and metastases. We excluded patients who were operated on for neural pathologies (e. g. gliomas). We recorded patient age, sex, the pathology for which the patient was operated on, and presence of edema in the biopsied brain specimen. We also recorded additional parameters including previous brain irradiation, past medical history, current medications, and monocyte counts and fractions. In the operating room, samples were collected from the cortex overlying the tumor. Sampled brain tissue was fixed in formaldehyde 4% (Sigma-Aldrich, HT501128) for 30 minutes, washed with phosphate buffered saline (PBS) and stored at 4°C in RNA*later*® (Merck, R0901-100M) until homogenization.

### NuNeX

Samples were homogenized with a Dounce tissue grinder (Sigma-Aldrich, D9063). The homogenate was passed through a 40 μm cell strainer (Corning, 352340) and pelleted by centrifugation at 900 g for 5 min. Cells were resuspended and stored in -80°C until FACS sorting.

### Immunostaining and FACS

Before cell sorting, Fc block (Invitrogen,14-9161-73) was added according to manufacturer's instructions at 20 μL/tube and incubated at 4°C for 20 min. For staining, anti-NeuN (1:500; Alexa Fluor®488-conjugated, Sigma, MAB377X) and anti-Iba1 (1:500; Abcam, ab178846) antibodies were added for 30 min incubation. Cells were then pelleted by centrifugation at 900 g for 5 min, resuspended in staining buffer, and stained with secondary antibody against anti-Iba1 (1:500; Alexa Fluor® 647- conjugated, Jackson ImmunoResearch, 711-605-152), followed by 30 min incubation. Re-pelleted cells were resuspended in staining buffer with DAPI (1:1000; Santa Cruz, sc-3598). All steps were performed on ice. Iba1- and NeuN-positive cells were sorted through a 85 μm nozzle with an approximate flow rate of 8,000 events/s. Sorted Iba1- and NeuN- positive cells were collected into tubes containing 500 μL staining buffer (> 1,000 cells), then centrifuged at 900 g for 5 min and resuspended in 100 μL of protein kinase digestion (PKD) buffer. RNA was extracted using a specialized kit (RNEasy® FFPE Kit, Qiagen, 73504). Samples were collected with BD FACSAria III (BD Biosciences) and analyzed with FCS Express 7 Software (De Novo Software).

### RNA sequencing and alignment of live human brain tissue-derived RNA

All samples underwent sequencing of small RNA. Libraries were constructed from 500 pg total RNA using D-Plex Small RNA-seq Kit for Illumina and Single Indexes for Illumina - Set B (Diagenode, C05030001). RIN after cell sorting was determined to be between 1.2 and 7.2 (Bioanalyzer 6000, Agilent) and the small RNA fraction was sequenced using the NextSeq 2000 System (Illumina) on the NextSeq 2000 System (Illumina) at the Center for Genomic Technologies Facility, the Hebrew University of Jerusalem. Small RNA was aligned to miRBase using miRexpress[77] and to the tRNA transcriptome using the MINTmap[78].

### ROSMAP subjects

RNA-Sequencing (RNA-Seq) data was derived from participants in the Rush Alzheimer's Disease Center's religious order study (ROS) and Rush memory and aging project (MAP) cohorts[63]. Both studies were approved by

an Institutional Review Board of Rush University Medical Center. All participants in ROSMAP enroll without known dementia and agree to annual clinical evaluation and brain donation at the time of death. Subjects are clinically classified for dementia, AD and Mild Cognitive Impairment (MCI) as previously reported[81,82]. Those without dementia or MCI were designated as no cognitive impairment (NCI)[83]. A summary diagnosis was made by a neurologist after death based on select clinical data blinded to all neuropathologic data. All cases received a neuropathological evaluation based upon Braak stage and CERAD which are combined to create NIA-Reagan pathologic criteria for AD[84]. Analysis shown in Fig. 2j includes 141 *postmortem* samples from the nucleus accumbens. The labels (AD and control) were assigned based on the summary diagnosis of AD and NCI. Persons with MCI were excluded. The summary metadata is provided in Supplementary Data 5.

### Differential Expression (DE) analysis
DE analysis was performed in R using the DESEQ2 package[66]. Pre-filtered raw counts of long RNAs and miRs included fragments with median expression above 10 counts per million (CPM). The raw counts of tRFs were normalized using the DESEQ2 median of ratios method in which counts are divided by sample-specific size factors determined by median ratio of gene counts relative to geometric mean per gene[66] and prefiltered to include fragments with median expression above 10 counts.

### Gene set enrichment
Gene set enrichment to identify biological processes was performed with the PANTHER tool of Gene Ontology resource[85]. The list of all fragments expressed above (median CPM $\geq$ 0) was used as a reference gene list.

### Cell type deconvolution
Cell type deconvolution was performed in python using the AUTOGENES package[22] (Aliee et al, 2021), using a single-cell dataset from whole murine brain as a reference[21].

### Elucidating AD- related RNA transcripts
Identifying AD-associated transcripts among DE long RNA was performed using the meta-analysis resource[26], combining lists of DE genes in 15 AD-APP mouse model studies and 12 AD-MAPT mouse model studies. The overlap of DE genes in Klotho KO with the DE genes in APP and MAPT models was calculated separately.

### Elucidating the protein-coding RNA transcripts changed at the protein level in AD
In order to identify transcripts encoding proteins with altered levels in AD (Fig. 1h,i), we used a proteomic dataset from a mouse 5XFAD model[31] and a meta-analysis of a series of human proteomic studies[32]. For mouse data, we took all the significantly changed proteins between 10-months-old 5XFAD mice and the age-matched wildtype mice (2961 proteins in total). For human data, we took all the proteins significantly altered in at least 5 bulk tissue proteomic studies (848 proteins in total).

### miR and tRF target prediction
Predicted miR mRNA targets were identified using miRNet[34]. Our selection included targets validated in the Mus musculus host organism by at least one of the techniques below: 5'RACE, ChIP-seq, ELISA, luciferase assay, FACS, GFP reporter assay, HITS-CLIP, microarray, immunoblot, immunocytochemistry, immunohistochemistry, immunoprecipitaion, LacZ reporter assay, qPCR, Western blot, Northern blot, quantitative proteomic approach, sequencing.
Predicted targeted tRFs were identified using tRFTar[42].

### Elucidating AD related miRs
AD-associated miRs were identified using the analysis of the short RNA-seq data derived from AD mouse models[38]. MiRs, changed with age or AD mutations, were identified using the Wilcoxon signed-rank test, applied to

log-transformed normalized counts. Fragments with $p_{adj} \leq 0.05$ after FDR correction were considered significant.

### Statistics and reproducibility
To establish the transcriptomic effect of Klotho knockout, the RNA profiles of 5 mouse WT brains were compared to 5 mouse brains with Klotho knockout. Differential expression was performed separately for polyadenylated RNA, microRNA and tRNA fragments using a DESEQ2 package[66]. The Gene Ontology analyses were performed with the PANTHER software[85,86] (https://geneontology.org). The deconvolution to cell types based on mRNA profiles was performed using the AUTOGENES package[22], and the yielded cell type proportions were compared between WT and Klotho-KO brains using the nonparametric Mann-Whitney ranked sum test. The Mann-Whitney test was also used to detect differences between individual microRNA and tRF expression profiles in the external datasets such as in comparison between WT and AD mouse models, CSF and brain samples of human AD patients and controls, as well as between tRF levels in control HEK293 and U2OS cell cultures and parallel cultures with Angiogenin knockout/overexpression. The statistical analysis of the pull-down assay mass spectrometry output was performed using a PERSEUS software[87]. To generate neuron- and microglia-specific short RNA profiles from live human brain tissue, 16 samples with neurons were compared with 16 microglia samples. Differential expression analysis was performed using the DESEQ2 package[66]. The linear correlation of the patient age with the levels of has-miR-335-5p and has-miR-150-5p was established using a Pearson correlation coefficient.

### Reporting summary
Further information on research design is available in the Nature Portfolio Reporting Summary linked to this article.

### Data availability
Raw aligned counts of polyadenylated RNA, microRNA and tRNA fragments from Klotho KO and WT mouse brains, as well as raw aligned counts of microRNA and tRNA fragments from neurons and microglia from live human brain tissue is available at https://doi.org/10.6084/m9.figshare.24100785. Differential expression results reported in the paper are provided in Supplementary Data.

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

## Acknowledgements

The authors thank the study participants of the Religious Orders Study (ROS) and Rush Memory and Aging Project (MAP) and staff of the Rush Alzheimer's Disease Center in Chicago. The authors also express their gratitude to Dr. Ola Karmi from the Research Infrastructures Center (Hebrew University of Jerusalem) for her guidance and assistance in FACS sorting, to Ms. Adi Tujerman from the Center for Genomic Technologies (Hebrew University of Jerusalem) for her assistance with RNA-seq experiments, to Dr. Hanan Schoffman from the Stein Family mass spectrometry center (Hebrew University of Jerusalem) for his assistance in Mass Spectrometry analysis, and Dr. Taylor Schmitz from the University of Western Ontario for inspiring discussions. This work was supported by the Israel Science Foundation (ISF; 1016/18; 3213/19 to H. Soreq), the National Institute of Health (NIH; Aging grant 5P01AG014449-21 to E. Mufson and H Soreq), Keter Holdings (to H. Soreq), Klogenix LLC (Boston, MA, USA), the Gatsby Charitable Foundation (to Y.L.), the K. Stein foundation (to H.S) and a joint grant by the Shaarei Zedek Medical center (to I. Paldor and H. Soreq). The ROS-MAP projects were supported by grants from the National Institute of Health (NIH; P30AG10161, P30AG72975, R01AG15819, R01AG17917, U01AG46152, U01AG61356). Serafima Dubnov is an awardee of PhD fellowships by Azrieli, Kaete-Klausner and TEVA foundations. The graphical abstracts in this article were created with BioRender.com.

## Author contributions

Conceptualization: S.D., I.P., C.R.A., H.S. Data collection: S.D., N.Y., O.Y., D.A.B., S.S., E.M., H.S. Data curation: S.D., H.S. Formal analysis: S.D. Funding acquisition: D.G., I.P., C.R.A., H.S. Methodology: S.D., E.R.B., N.Y., O.Y., Y.T., M.K. Project administration: S.D., H.S. Supervision: D.G., C.R.A., H.S. Visualization: S.D. Writing – original draft: S.D., C.R.A., H.S. Writing – review and editing: S.D., E.R.B., H.S.

## Competing interests

The authors declare no competing interests.
