## [Peer Review File · Communications Biology]

Reviewers' comments:

Reviewer #1 (Remarks to the Author):

The manuscript by Dubnov et al. entitled “Knockout of the longevity gene Klotho perturbs aging- and Alzheimer’s disease-linked brain microRNAs and tRNA fragments” describes how KO of Klotho alters expression on long RNA, miRNA, and tRNA in mouse brains and attempts to relate these findings to Alzheimer’s disease and aging. This paper is interesting as Klothos is a gene of interest in many diseases and understanding its regulatory networks may assist in the identification of therapeutic targets. However, this paper is not suitable for publication in its current format. My major concern is that it is missing critical information required to properly interpret the results. Additionally, clarity should be improved at some points detailed below.

Major concerns

1. The authors refer to poly(A)-selected RNA as “long RNA”. I would like to request the authors be more specific about the specific population of RNA they are sequencing. It appears that the “long RNA” they are sequencing is mRNA, in which case it should be referred to as mRNA not long RNA to distinguish it from other “long RNAs” (e.g. lncRNA).
2. “Identifying DE miRs targets using miRNet30 revealed the experimentally validated targets whose levels were repressed by the corresponding miRs in other experimental systems.” Which targets and which experimental systems? Are the experimental systems relevant to this body of work? Please expand.
3. When comparing the NuNeX data to the mouse data please indicate how many of the mmu-miR have human orthologs.
4. The discussion is very short and does little to place this findings in the context of the wider field.
5. There is some missing information related to the mouse strain that should be included in the methods. This includes sex, age, and genetic background. Additionally, a reference should also be provided for the mice (either a paper characterising the KO or if commercially available a catalogue number would be sufficient).
6. Different extraction and sequencing techniques were used throughout the paper, most notably whole cell vs. NuNeX. Different techniques affect the RNA that are present. NuNeX is nuclear enriched with “some associated cytoplasm” and it is well know that different subsets of RNAs, including miRNA, are detected in nuclear vs. whole cell sequencing. The authors should discuss how the use of different techniques may impact their results.
7. Little information has been provided regarding the patient demographics from the “live human brain samples” despite the authors stating they have recorded a wide variety of

parameters in the “tissue sampling” section of the methods. A table of this information should be included and list more of these parameters than is stated briefly in the results (6 females, 10 males; 42-75 years old).

Minor concerns:

1. The abbreviation list is incomplete (e.g. mmu-miR is included but hsa-miR is not, LTP is not included)

Reviewer #2 (Remarks to the Author):

Summary:

The submitted manuscript by Dunov and colleagues reports that multiple Klotho deficiency perturbed aging- and neurodegeneration -related long and short RNA transcripts in both neurons and glia from the murine and human brain.

Strengths:

Overall, the work is focused and utilizes several state-of-the-art techniques. Results are well described and the language is up to the mark. The results show what the authors have claimed. -Figures are clear, eye-catching, and professionally prepared- something Nature or other journals love to see.

All these points are well expected from a highly resource-rich and large group of investigators from different labs.

Weaknesses:

The use of live human brain samples is one of the major strengths; however, diversity of conditions dampens the enthusiasm. For example, the authors claimed non-neural pathologies, including meningiomas, metastases, and vascular malformations; however, the “vascular” component is increasingly believed to be part of AD pathology, and how could they be labeled as “non-neural pathologies.”

What is not clear is the implication of such sophisticated work. What did happen to protein levels as a consequence of changes in miR and tRNA fragments? After all, it is the protein that does the job in neurodegenerative diseases and AD. The authors are aware of the role of proteins, as they mentioned "proteins" extensively in the Intro (not in the Results). Data on protein levels would have strengthened significantly the manuscript.

Response to reviewers

The reviewers' comments are provided in black, while the authors' response – in red.

Reviewer #1 (Remarks to the Author):

The manuscript by Dubnov et al. entitled “Knockout of the longevity gene Klotho perturbs aging- and Alzheimer’s disease-linked brain microRNAs and tRNA fragments” describes how KO of Klotho alters expression on long RNA, miRNA, and tRNA in mouse brains and attempts to relate these findings to Alzheimer’s disease and aging. This paper is interesting as Klotho is a gene of interest in many diseases and understanding its regulatory networks may assist in the identification of therapeutic targets. However, this paper is not suitable for publication in its current format. My major concern is that it is missing critical information required to properly interpret the results. Additionally, clarity should be improved at some points detailed below.

Major concerns

1. The authors refer to poly(A)-selected RNA as “long RNA”. I would like to request the authors be more specific about the specific population of RNA they are sequencing. It appears that the “long RNA” they are sequencing is mRNA, in which case it should be referred to as mRNA not long RNA to distinguish it from other “long RNAs” (e.g. lncRNA).

Thank you for this important comment and for noting our error. Indeed, we referred to polyadenylated RNA as “long RNA”, but our sequenced transcripts included both mRNA and long non-coding RNAs, most of which are polyadenylated as well. We have now revised the text accordingly and included an additional analysis of the differentially expressed long non-coding RNAs included in the dataset. The outcome results are detailed in the new Supplementary Figure 2, provided below, and explained in the text in yellow highlighted additions (lines 154-159).

Supplementary Figure 2: Long non-coding RNA differentially expressed in Klotho KO

A. PCA based on long non-coding RNA reads colored by genotype. **B.** Heatmap showing normalized and centered counts of the significant long non-coding RNAs

2. “Identifying DE miRs targets using miRNet30 revealed the experimentally validated targets whose levels were repressed by the corresponding miRs in other experimental

systems.” Which targets and which experimental systems? Are the experimental systems relevant to this body of work? Please expand.

Our apologies on this unclear text, which indeed required expansion. Briefly, the targets referred to are now listed in Supplementary Table 3. In addition, we expanded the text in the Methods section and presented the experimental system in a longer and hopefully clearer text. Finally, we reported in the Methods section all of those identified miR targets which had been experimentally validated in the *Mus Musculus* host organism. All of these additional text and Table are highlighted in yellow in the revised manuscript (lines 715-721).

3. When comparing the NuNeX data to the mouse data please indicate how many of the mmu-miR have human orthologs.

This is indeed a most interesting point and we appreciate the opportunity to refer to it in our revised manuscript. As indicated by the closed similarity in the impact of *Klotho* in mice and human brains, we found many orthologues of the murine miRs in human data sources. To highlight this topic in our revised manuscript, we counted the number of shared miRs between mouse and human annotation datasets and added information about these orthologs in the revised text (lines 424-427).

4. The discussion is very short and does little to place these findings in the context of the wider field.

Indeed, our discussion has been too brief, largely due to our respect for the brief text requests by the journal. Given your revision request, we have now expanded the discussion and added an explanation of the novelty of our findings in the context of the wider research field. The main message of this wider discussion indeed refers to the novelty of our findings, which expand the issue of *Klotho*'s impact on the mammalian brain to the realm of Alzheimer's disease and in particular adds the impact of transfer RNA fragments (tRFs) on *Klotho*'s longevity effects (lines 486-505).

5. There is some missing information related to the mouse strain that should be included in the methods. This includes sex, age, and genetic background. Additionally, a reference should also be provided for the mice (either a paper characterising the KO or if commercially available a catalogue number would be sufficient).

Our apologies for not including these details before. Information about the sex, age and genetic background of the studied mice has now been provided in the revised Methods section, and the original paper based on *Klotho*'s impact in the same mouse model has now been cited (lines 538-540).

6. Different extraction and sequencing techniques were used throughout the paper, most notably whole cell vs. NuNeX. Different techniques affect the RNA that are present. NuNeX is nuclear enriched with “some associated cytoplasm” and it is well known that different subsets of RNAs, including miRNA, are detected in nuclear vs. whole cell sequencing. The authors should discuss how the use of different techniques may impact their results.

Many thanks for this comment, which we have found to be most justified. Indeed, our NuNeX protocol enables isolation of single nuclei surrounded by small amounts of

cytoplasm which makes it different from both single nuclei and single cell RNA-sequencing protocols. Our revised discussion refers to the unique pattern of such isolated nuclei as a technical limitation which however enables rapid and specific collection of such cytoplasm-surrounded single nuclei of particular immune-labeled cell types and their FACS selection from live human brains (lines 514-517).

7. Little information has been provided regarding the patient demographics from the “live human brain samples” despite the authors stating they have recorded a wide variety of parameters in the “tissue sampling” section of the methods. A table of this information should be included and list more of these parameters than is stated briefly in the results (6 females, 10 males; 42-75 years old).

These requests are fully justified and important, thanks for addressing this issue. In response, we have now added a new table with the full medical information as received from our clinical co-authors, as Supplementary Tables 9, referring to it in the text in lines 408-410.

Minor concerns:

1. The abbreviation list is incomplete (e.g. mmu-miR is included but hsa-miR is not, LTP is not included)

Sorry about that, we have now extended the abbreviation list to comply with this additional request.

Reviewer #2 (Remarks to the Author):

Summary:

The submitted manuscript by Dubnov and colleagues reports multiple Klotho deficiency perturbed aging- and neurodegeneration -related long and short RNA transcripts in both neurons and glia from the murine and human brain.

Strengths:

Overall, the work is focused and utilizes several state-of-the-art techniques. Results are well described and the language is up to the mark. The results show what the authors have claimed. -Figures are clear, eye-catching, and professionally prepared- something Nature or other journals love to see.

All these points are well expected from a highly resource-rich and large group of investigators from different labs.

Weaknesses:

The use of live human brain samples is one of the major strengths; however, diversity of conditions dampens the enthusiasm. For example, the authors claimed non-neural pathologies, including meningiomas, metastases, and vascular malformations; however, the “vascular” component is increasingly believed to be part of AD pathology, and how could they be labeled as “non-neural pathologies.”

Many thanks for your largely positive view and apologies for the weaker parts. Indeed, we pay a high price for getting access to live human brain tissues, in that none of the

samples were drawn from fully healthy patients- simply since they are not subjected to neurosurgery. To fully refer to this limitation as implicated from your comment, we have now added more explanations and details to the revised manuscript. Briefly, the samples included in the current manuscript were all derived from surgical removal of tumor metastases and meningioma pathologies, but did **not include any** vascular deformations. We corrected the text in the methods section to highlight these issues (lines 628-630).

What is not clear is the implication of such sophisticated work. What did happen to protein levels as a consequence of changes in miR and tRNA fragments? After all, it is the protein that does the job in neurodegenerative diseases and AD. The authors are aware of the role of proteins, as they mentioned "proteins" extensively in the Intro (not in the Results). Data on protein levels would have strengthened significantly the manuscript.

Thank you for this most relevant comment. Following your advice, we validated parallel changes in the protein products to those of the initially reported mRNA alterations of the brain mRNAs. Briefly, we employed Alzheimer's disease proteomic datasets, both from mouse and human brains and were happy to validate the transcript changes at the level of proteins. As for the Klotho KO brain tissues, those have been flown in from Japan, and have all been used in our previous work. No tissue was left of those samples, and international flights to Israel are problematic at present, which limited our handling of this matter to the confirmatory dataset analysis referred to above. The added sentences are yellow-highlighted in the revised manuscript (lines 173-186, 707-713), and the corresponding results are reported in newly added Figure 1H and Figure 1I. The updated Figure 1 is provided below.

Figure 1. Klotho KO induces numerous long RNA transcriptome changes.

A. Graphical representation of the analysis shown in Figure 1. **B.** Volcano plot of polyadenylated RNAs DE in murine Klotho-knockout compared to wildtype brains; upregulated transcripts in blue, downregulated transcripts in pink and; $p_{adj} \leq 0.05$. **C.** PCA of polyadenylated RNA reads colored by genotype. **D.** Heatmap showing normalized and centered counts of the most significantly altered transcripts ($p_{adj} \leq 5e-6$; colormap - z-score of normalized count levels, centered to the mean). **E.** GO annotation of biological processes enriched in up- and down-regulated transcripts (blue and pink, respectively) under Klotho-knockout. **F.** AutoGeneS prediction of the cell type proportions in WT and KO profiles based on the scRNA-seq atlas²¹; **: $p_{val} < 0.01$, ns: non-significant. **G.** Pie chart proportions of DE genes in AD studies²⁶. **H.** Bar plot showing the percent of DE mRNAs encoding proteins that are altered in a mouse model AD study³¹, in a compilation of human AD proteomic studies³² and in both. **I.** GO annotation of molecular functions enriched in protein-coding transcripts whose associated proteins were altered in a compilation of human AD proteomic studies³².

A**B****C****D****E****G****H****F****I**
In addition, we developed and established a novel experimental protocol whereby the proteins bound to an examined transfer RNA fragment (tRF) of interest are mapped and validated, and the corresponding networks are established. Briefly, this approach has been based on a pull-down assay in which we exposed the tRF upregulated under Klotho knockout to neuronal proteins, washed the mixture carefully and analyzed its contents by mass spectrometry followed by in-depth statistical analysis. We were happy to learn that the tested tRF is involved in a protein network of alternative splicing and RNA regulation. Specifically, this protocol revealed that the identified tRF of interest interacts with the spliceosome and with proteins involved in mRNA decay and chemical modification processes, attributing to this tRF pivotal brain activities related to Klotho's knockout. This new results section is highlighted in yellow (lines 324-375, 560-620) and summarized in the new Figure 4, provided below.

Figure 4. Pull-down assay revealing that tDR-36:74-Asn-GTT-2-M2 binds multiple proteins involved in splicing, mRNA decay and pseudouridylation.

A. Graphical representation of the pull-down experiment: SH-SY5Y cells were plated, harvested after four days, and incubated with magnetic streptavidin beads to which biotinylated tDR-36:74-Asn-GTT-2-M2 (Klotho-tRF) or biotinylated negative control (NC) RNA-oligonucleotides were bound. After wash steps the beads were submitted to mass spectrometry and the proteins bound to each oligonucleotide were identified. **B.** PCA showing separation of the proteins bound to the tDR-36:74-Asn-GTT-2-M2 (purple) vs. those bound to the NC oligo (green), 4 samples per group. **C.** Volcano plot showing the differentially expressed proteins in biotinylated tDR-36:74-Asn-GTT-2-M2 vs biotinylated NC pulldown samples. Proteins upregulated with $\log_2FC \geq 4$ are colored in purple, proteins downregulated with $\log_2FC \leq -4$ are colored in green. The statistical analysis was performed using the Perseus package. **D.** List of the 13 proteins enriched by $\log_2FC \geq 4$ (16-fold) in tDR-36:74-Asn-GTT-2-M2 vs. NC samples. **E.** Gene Ontology annotation of the biological processes enriched within the differentially expressed proteins in panel D.

D

Gene names	log ₂ (FC)	P-value
CNBP	8.0	2.6E-05
GRSF1	7.5	5.3E-05
HNRNPF	7.1	4.0E-06
SART3	6.1	2.6E-05
SKIV2L	5.6	4.1E-05
HNRNPH3	5.4	1.7E-06
SNRPA	5.0	1.1E-03
SNRNP70	4.7	5.4E-07
NCL	4.5	1.4E-08
HNRNPH1	4.3	7.3E-10
TTC37	4.3	4.5E-05
TRUB1	4.1	1.4E-04
PUS7	4.1	7.0E-05

E

REVIEWERS' COMMENTS:

Reviewer #1 (Remarks to the Author):

The manuscript by Dubnov et al. entitled "Knockout of the longevity gene Klotho perturbs aging- and Alzheimer's disease-linked brain microRNAs and tRNA fragments" describes how KO of Klotho alters expression on long RNA, miRNA, and tRNA in mouse brains and attempts to relate these findings to Alzheimer's disease and aging. This paper is interesting as Klothos is a gene of interest in many diseases and understanding its regulatory networks may assist in the identification of therapeutic targets. My major concern with this paper was several instances of missing critical information required to properly interpret the results. The authors have successfully addressed these issues. I have no further concerns with this paper.

Reviewer #2 (Remarks to the Author):

Response to the reviewers' comments is partial and somewhat satisfactory.